# Learning Hawkes Processes from a Handful of Events

**Farnood Salehi**[*]
EPFL
farnood.salehi@epfl.ch

**William Trouleau**[*]
EPFL
william.trouleau@epfl.ch

**Matthias Grossglauser**
EPFL
matthias.grossglauser@epfl.ch

**Patrick Thiran**
EPFL
patrick.thiran@epfl.ch

## Abstract

Learning the causal-interaction network of multivariate Hawkes processes is a useful task in many applications. Maximum-likelihood estimation is the most common approach to solve the problem in the presence of long observation sequences. However, when only short sequences are available, the lack of data amplifies the risk of overfitting and regularization becomes critical. Due to the challenges of hyper-parameter tuning, state-of-the-art methods only parameterize regularizers by a single shared hyper-parameter, hence limiting the power of representation of the model. To solve both issues, we develop in this work an efficient algorithm based on variational expectation-maximization. Our approach is able to optimize over an extended set of hyper-parameters. It is also able to take into account the uncertainty in the model parameters by learning a posterior distribution over them. Experimental results on both synthetic and real datasets show that our approach significantly outperforms state-of-the-art methods under short observation sequences.

## 1 Introduction

In many real-world applications, including finance, computational biology, social-network studies, criminology, and epidemiology, it is important to gain insight from the interactions of multivariate time series of discrete events. For example, in finance, changes in the price of a stock might affect the market [4]; and in epidemiology, individuals infected by an infectious disease might spread the disease to their neighbors [15]. Such networks of time series often exhibit mutually exciting patterns of diffusion. Hence, a recurring issue is to learn in an unsupervised way the causal structure of interacting networks. This task is usually tackled by defining a so-called causal graph of entities where an edge from a node $i$ to a node $j$ means that events in node $j$ depend on the history of node $i$. Such causal interactions are typically learned with either directed information [24, 23], transfer entropy [26], or Granger causality [2, 11].

A widely used model for capturing mutually exciting patterns in a multi-dimensional time series is the Multivariate Hawkes process (MHP), a particular type of temporal point process where an event in one dimension can affect future arrivals in other dimensions. It has been shown that learning the excitation matrix of an MHP encodes the causal structure between the processes, both in terms of Granger causality [12] and directed information [13]. Most studies focus on the scalability of MHPs to large datasets. However, in many applications, data can be very expensive to collect, or simply not available. For example, in economic and public health studies, collecting survey data is usually an expensive process. Similarly, in the case of epidemic modeling, it is critical to learn as fast as possible the patterns of diffusion of a spreading disease. As a result, the amount of data

---

[*]The first two authors contributed equally to this work.

available is intrinsically limited. MHPs are known to be sensitive to the amount of data used for training, and the excitation patterns learned by MHPs from short sequences can be unreliable [29]. In such settings, the likelihood becomes noisy and regularization becomes critical. Nevertheless, as most hyper-parameter tuning algorithms such as grid search, random search, and even Bayesian optimization become challenging when the number of hyper-parameters is large, state-of-the-art methods only parameterize regularizers by a single shared hyper-parameter, hence limiting the power of representation of the model.

In this work, we address both the small data and hyper-parameter tuning issues by considering the parameters of the model as latent variables and by developing an efficient algorithm based on variational expectation-maximization. By estimating the evidence – rather than the likelihood – the proposed approach is able to optimize, with minimal computational complexity, over an extended set of hyper-parameters. Our approach is also able to take into account the uncertainty in the model parameters by fitting a posterior distribution over them. Therefore, rather than just providing a point estimate, this approach can provide an estimation of uncertainty on the learned causal graph. Experimental results on synthetic and real datasets show that, as a result, the proposed approach significantly outperforms state-of-the-art methods under short observation sequences, and maintains the same performance in the large-data regime.

## 2   Related Works

The most common approaches to learning the excitation matrix of MHPs are based on variants of regularized maximum-likelihood estimation (MLE). Zhou et al. [31] propose regularizers that enforce sparse and low-rank structures, along with an efficient algorithm based on the alternating-direction method of multipliers. To mitigate the parametric assumption, Xu et al. [28] represent the excitation functions as a series of basis functions, and to achieve sparsity under this representation they propose a sparse group-lasso regularizer. Such estimation methods are often referred to as non-parametric as they enable more flexibility on the shape of the excitation functions [19, 16]. To estimate the excitation matrix without any parametric modeling, fully non-parametric approaches were developed [32, 1]. However, these methods focus on scalability and target settings where large-scale datasets are available.

Bayesian methods go beyond the classic approach of MLE by enabling a probabilistic interpretation of the model parameters. A few studies tackled the problem of learning MHPs from a Bayesian perspective. Linderman and Adams [20] use a Gibbs sampling-based approach, but the convergence of the proposed algorithm is slow. To tackle this problem, Linderman and Adams [21] discretize the time, which introduces noise in the model. In a different setting where some of the events or dimensions are hidden, Linderman et al. [22] use an expectation maximization algorithm to marginalize over the unseen part of the network.

Bayesian probabilistic models are usually intractable and require approximate inference. To address the issue, variational inference (VI) approximates the high-dimensional posterior of the probabilistic model. It recently gained interest in many applications. VI is used, to name a few, for word embedding [5, 7], paragraph embedding [17], and knowledge-graph embedding [6]. For more details on this topic, we refer the reader to Zhang et al. [30] and Blei et al. [9]. Variational inference has also proven to be a successful approach to learning hyper-parameters [8, 6]. Building on recent advances in variational inference, we develop in this work a variational expectation-maximization algorithm by interpreting the parameters of an MHP as latent variables of a probabilistic model.

## 3   Preliminary Definitions

### 3.1   Multivariate Hawkes Processes

Formally, a $D$-dimensional MHP is a collection of $D$ univariate counting processes $N_i(t)$, $i = 1, \ldots, D$, whose realization over an observation period $[0, T)$ consists of a sequence of discrete events $\mathcal{S} = \{(t_n, i_n)\}$, where $t_n \in [0, T)$ is the timestamp of the $n$-th event and $i_n \in \{1, \ldots, D\}$ is

its dimension. Each process has the particular form of conditional intensity function

$$\lambda_i(t) = \mu_i + \sum_{j=1}^{D} \int_0^t \phi_{ij}(t - \tau)dN_j(\tau), \tag{1}$$

where $\mu_i > 0$ is the constant exogenous part of the intensity of process $i$, and the excitation function $\phi_{ij} : \mathbb{R}_+ \mapsto \mathbb{R}_+$ encodes the effect of past events from dimension $j$ on future events from dimension $i$. The larger the values of $\phi_{ij}(t)$, the more likely events in dimension $j$ will trigger events in dimension $i$. It has been shown that the excitation matrix $[\phi_{ij}(t)]$ encodes the causal structure of the MHP in terms of Granger causality, $i.e.$, $\phi_{ij}(t) = 0$ if and only if the process $j$ does not Granger-cause process $i$ [13, 12].

Most of the literature uses a parametric form for the excitation functions. The most popular form is the exponential excitation function

$$\phi_{ij}(t) = w_{ij}\zeta e^{-\zeta t}. \tag{2}$$

However, in most applications the excitation patterns are unknown and this form might be too restrictive. Hence, to alleviate the assumption of a particular form for the excitation function, other approaches [19, 16, 28] over-parameterize the space and encode the excitation functions as a linear combination of $M$ basis functions $\kappa_1(t), \kappa_2(t), \ldots, \kappa_M(t)$ as

$$\phi_{ij}(t) = \sum_{m=1}^{M} w_{ij}^m \kappa_m(t), \tag{3}$$

where the basis functions are often exponential or Gaussian kernels [28]. This kind of approach is generally referred to as non-parametric. In the experiments of Section 5, we investigate the performance of both parametric and non-parametric approaches to learning MHPs from small sequences of observations. We denote the set of exogenous rates by $\boldsymbol{\mu} = \{\mu_i\}_{i=1}^{D} \in \mathbb{R}_+^D$ and the excitation matrix by $\boldsymbol{W} := \{\{w_{ij}^m\}_{m=1}^{M}\}_{i,j=1}^{D} \in \mathbb{R}_+^{D^2M}$.

## 3.2 Maximum Likelihood Estimation

Suppose that we observe a sequence of discrete events $\mathcal{S} = \{(t_n, i_n)\}_{n=1}^{N}$ over an observation period $[0, T)$. The most common approach to learning the parameters of an MHP given $\mathcal{S}$ is to perform a regularized maximum-likelihood estimation [31, 3, 28], which amounts to minimizing an objective function that is the sum of the negative log-likelihood and of a penalty term that induces some desired structural properties. Specifically, the objective is to solve the optimization problem

$$\hat{\boldsymbol{\mu}}, \hat{\boldsymbol{W}} = \underset{\boldsymbol{\mu} \geq 0, \boldsymbol{W} \geq 0}{\operatorname{argmin}} - \log p(\mathcal{S}|\boldsymbol{\mu}, \boldsymbol{W}) + \frac{1}{\alpha}\mathcal{R}(\boldsymbol{\mu}, \boldsymbol{W}), \tag{4}$$

where the log-likelihood of the parameters is given by

$$\log p(\mathcal{S}|\boldsymbol{\mu}, \boldsymbol{W}) = \sum_{(t_n, i_n) \in \mathcal{S}} \log \lambda_{i_n}(t_n) - \sum_{i=1}^{D} \int_0^T \lambda_i(t)dt. \tag{5}$$

The particular choice of penalty $\mathcal{R}(\boldsymbol{\mu}, \boldsymbol{W})$, along with the single hyper-parameter $\alpha$ controlling its influence, depends on the problem at hand. For example, a necessary condition to ensure that the learned model is stable is that $\lim_{t \to \infty} \phi_{ij}(t) = 0$ and that the spectral radius of the excitation matrix is less than 1 [10]. Hence, a common penalty used is

$$\mathcal{R}_p(\boldsymbol{\mu}, \boldsymbol{W}) = \sum_{i,j=1}^{d} \sum_{m=1}^{M} |w_{ij}^m|^p, \tag{6}$$

with $p = 1$ or 2 in [32, 31, 28]. Another common assumption is that the graph is sparse. In this case, a Group-Lasso penalty of the form

$$\mathcal{R}_{1,2}(\boldsymbol{\mu}, \boldsymbol{W}) = \sum_{i,j=1}^{d} \sqrt{\sum_{m=1}^{M} (w_{ij}^m)^2} \tag{7}$$

is commonly used to enforce sparsity in the excitation functions [28].

Small data amplifies the danger of overfitting; hence the choice of regularizers and their hyper-parameters becomes essential. Nevertheless, to control the influence of the penalty in (4), all state-of-the-art methods are limited by the use of a single shared hyper-parameter $\alpha$. Ideally, we would have a different hyper-parameter to independently control the effect of the penalty on each parameter of the model. However, the number of parameters, *i.e.*, $(D^2 M + D)$, grows quadratically with the dimension of the problem $D$. To make matters worse, the most common approaches used to fine-tune the choice of hyper-parameters, *i.e.*, grid search and random search, become computationally prohibitive when the number of hyper-parameters becomes large. Indeed, the search space exponentially increases with the number of hyper-parameters. Another approach is to use Bayesian optimization of hyper-parameters, but the cost of doing this also becomes prohibitive as the number of samples required to learn the landscape of cost function exponentially increases with the number of hyper-parameters [27]. We describe the details of our proposed approach in the next section.

## 4 Proposed Learning Approach

We now introduce the proposed approach for learning MHPs. The approach enables us to use different hyper-parameters for each model parameter and efficiently tune them all by taking into account parameter uncertainty. It is based on the variational expectation-maximization (EM) algorithm and jointly optimizes both the model parameters $\boldsymbol{\mu}$ and $\boldsymbol{W}$, as well as the hyper-parameters $\boldsymbol{\alpha}$.

First, we can view regularized MLE as a maximum a posteriori (MAP) estimator of the model where parameters are considered as latent variables. Under this interpretation, regularizers on the model parameters correspond to unnormalized priors on the latent variables. The optimization problem becomes

$$\hat{\boldsymbol{\mu}}, \hat{\boldsymbol{W}} = \underset{\boldsymbol{\mu} \geq 0, \boldsymbol{W} \geq 0}{\operatorname{argmax}} \log p_{\boldsymbol{\alpha}}(\boldsymbol{\mu}, \boldsymbol{W}, \mathcal{S}) = \underset{\boldsymbol{\mu} \geq 0, \boldsymbol{W} \geq 0}{\operatorname{argmax}} \log p(\mathcal{S}|\boldsymbol{\mu}, \boldsymbol{W}) + \log p_{\boldsymbol{\alpha}}(\boldsymbol{\mu}, \boldsymbol{W}). \quad (8)$$

Therefore, having a better regularizer means having a better prior. In the presence of a long sequence of observations, we want the prior to be as uninformative as possible (*i.e.*, a smaller regularization) as we have access to enough information for the MLE to accurately estimate the parameters of the model. But in the case where we only observe short sequences, we want to use more informative priors to avoid overfitting (*i.e.*, a larger regularization).

Unfortunately, the MAP estimator cannot adjust the influence of the prior by optimizing over $\boldsymbol{\alpha}$. Indeed, the cost function in (8) is unbounded from above and solving Equation (8) with respect to $\boldsymbol{\alpha}$ leads trivially to a divergent solution $\frac{1}{\alpha} \to \infty$. To address this issue, we can take a Bayesian approach, integrate out parameters and optimize the evidence (or marginal likelihood) $p_{\boldsymbol{\alpha}}(\mathcal{S})$ instead of the log-likelihood. Such an approach changes the optimization problem of Equation (8) into

$$\hat{\boldsymbol{\alpha}} = \underset{\boldsymbol{\alpha} \geq 0}{\operatorname{argmax}} \iint p(\mathcal{S}|\boldsymbol{\mu}, \boldsymbol{W}) p_{\boldsymbol{\alpha}}(\boldsymbol{\mu}, \boldsymbol{W}) d\boldsymbol{\mu} d\boldsymbol{W} = \underset{\boldsymbol{\alpha} \geq 0}{\operatorname{argmax}} \ p_{\boldsymbol{\alpha}}(\mathcal{S}). \quad (9)$$

Unlike the MAP objective function, maximizing the evidence over $\boldsymbol{\alpha}$ does not lead to a degenerate solution because it is upper bounded by the likelihood. However, this optimization problem can be solved only for simple models where the integral has a closed form, which requires a conjugate prior to the likelihood. Therefore, we use variational inference to estimate the evidence and develop a variational EM algorithm to optimize our objective with respect to $\boldsymbol{\alpha}$.

### 4.1 Variational Expectation-Maximization for Multivariate Hawkes Processes

**Variational inference.** The derivation of the variational objective is as follows. First postulate a variational distribution $q_{\boldsymbol{\gamma}}(\boldsymbol{\mu}, \boldsymbol{W})$, parameterized by the variational parameters $\boldsymbol{\gamma}$, approximating the posterior $p(\boldsymbol{\mu}, \boldsymbol{W}|\mathcal{S})$. The variational parameters $\boldsymbol{\gamma}$ are chosen such that the Kullback–Leibler divergence between the true posterior $p(\boldsymbol{\mu}, \boldsymbol{W}|\mathcal{S})$ and the variational distribution $q_{\boldsymbol{\gamma}}(\boldsymbol{\mu}, \boldsymbol{W})$ is minimized. It is known that minimizing $KL\left[q_{\boldsymbol{\gamma}}(\boldsymbol{\mu}, \boldsymbol{W}) \| p(\boldsymbol{\mu}, \boldsymbol{W}|\mathcal{S})\right]$ is equivalent to maximizing the *evidence lower-bound* (ELBO) [9, 30] defined as

$$\mathsf{ELBO}(q_{\boldsymbol{\gamma}}, \boldsymbol{\alpha}) := \mathbb{E}_{q_{\boldsymbol{\gamma}}}\left[\log p_{\boldsymbol{\alpha}}(\boldsymbol{\mu}, \boldsymbol{W}, \mathcal{S})\right] - \mathbb{E}_{q_{\boldsymbol{\gamma}}}[\log q_{\boldsymbol{\gamma}}(\boldsymbol{\mu}, \boldsymbol{W})]. \quad (10)$$

By invoking Jensen's inequality on the integral $p_{\boldsymbol{\alpha}}(\mathcal{S}) = \iint p_{\boldsymbol{\alpha}}(\boldsymbol{\mu}, \boldsymbol{W}, \mathcal{S}) d\boldsymbol{\mu} d\boldsymbol{W}$, we obtain the desired lower bound on the evidence $p_{\boldsymbol{\alpha}}(\mathcal{S}) \geq \mathsf{ELBO}(q_{\boldsymbol{\gamma}}, \boldsymbol{\alpha})$ where, by maximizing $\mathsf{ELBO}(q_{\boldsymbol{\gamma}}, \boldsymbol{\alpha})$ with respect to $\boldsymbol{\gamma}$, the bound becomes tighter.

For simplicity, we adopt the mean-field assumption by choosing a variational distribution $q_{\boldsymbol{\gamma}}(\boldsymbol{\mu}, \boldsymbol{W})$ that factorizes over the latent variables[2]. As the parameters $\boldsymbol{\mu}$ and $\boldsymbol{W}$ of an MHP are non-negative, a good candidate to approximate the posterior is a log-normal distribution. We define the variational parameters $\boldsymbol{\gamma} = \{\boldsymbol{\nu}, e^{\boldsymbol{\sigma}}\}$ as the mean and the standard deviation of $q_{\boldsymbol{\gamma}}$. We denote the standard deviation by $e^{\boldsymbol{\sigma}}$ because, we optimize its log to naturally ensure its positivity and the stability of the optimization procedure. Although we present our learning approach for the log-normal distribution, it is easily generalizable to other distributions.

**Variational EM algorithm.** In order to efficiently optimize the ELBO with respect to both the variational parameters $\boldsymbol{\gamma}$ and the hyper-parameters $\boldsymbol{\alpha}$, we use the variational EM algorithm that iterates over the two following steps: The E-step maximizes the ELBO with respect to the variational parameters $\boldsymbol{\gamma}$ in order to get a tighter lower-bound on the evidence; and the M-step updates the hyper-parameters $\alpha$ with a closed form update. Details of the two steps are as follows.

The E-step maximizes the ELBO with respect to the variational parameters $\boldsymbol{\gamma}$ to make the variational distribution $q_{\boldsymbol{\gamma}}(\boldsymbol{\mu}, \boldsymbol{W})$ close to the exact posterior $p(\boldsymbol{\mu}, \boldsymbol{W}|\mathcal{S})$ and to ensure that the ELBO is a good proxy for the evidence. To evaluate the ELBO, we use the black-box variational-inference optimization in [18, 25]. Re-parameterize the model as

$$\boldsymbol{\mu} = g_{\boldsymbol{\gamma}}(\boldsymbol{\varepsilon}^{\boldsymbol{\mu}}) = \exp(\boldsymbol{\nu}^{\boldsymbol{\mu}} + e^{\boldsymbol{\sigma}^{\boldsymbol{\mu}}} \odot \boldsymbol{\varepsilon}^{\boldsymbol{\mu}}),$$

$$\boldsymbol{W} = g_{\boldsymbol{\gamma}}(\boldsymbol{\varepsilon}^{\boldsymbol{W}}) = \exp(\boldsymbol{\nu}^{\boldsymbol{W}} + e^{\boldsymbol{\sigma}^{\boldsymbol{W}}} \odot \boldsymbol{\varepsilon}^{\boldsymbol{W}}),$$

where $\boldsymbol{\varepsilon}^{\boldsymbol{\mu}}$ (resp. $\boldsymbol{\varepsilon}^{\boldsymbol{W}}$) has the same shape as $\boldsymbol{\mu}$ (resp. $\boldsymbol{W}$), with each element following a normal distribution $\mathcal{N}(0, 1)$. $\odot$ denotes the element-wise product. This trick enables us to rewrite the first intractable expectation term of the ELBO in (10) as

$$\mathbb{E}_{q_{\boldsymbol{\gamma}}}\left[\log p_C(\boldsymbol{\mu}, \boldsymbol{W}, \mathcal{S})\right] = \mathbb{E}_{\boldsymbol{\varepsilon} \sim \mathcal{N}(0, I)}\left[\log p_{\boldsymbol{\alpha}}\left(g_{\boldsymbol{\gamma}}(\boldsymbol{\varepsilon}^{\boldsymbol{\mu}}), g_{\boldsymbol{\gamma}}(\boldsymbol{\varepsilon}^{\boldsymbol{W}}), \mathcal{S}\right)\right]. \tag{11}$$

The second term of the ELBO in (10) is the entropy of the log-normal distribution that can be expressed, up to a constant, as $\sum_{\mu_i, \sigma_i}(\mu_i + \sigma_i)$. The ELBO then can be estimated by Monte-Carlo integration as

$$\text{ELBO}(q_{\boldsymbol{\gamma}}, \boldsymbol{\alpha}) \approx \frac{1}{L} \sum_{\ell=1}^{L} \log p_{\boldsymbol{\alpha}}\left(g_{\boldsymbol{\gamma}}(\boldsymbol{\varepsilon}_{\ell}^{\boldsymbol{\mu}}), g_{\boldsymbol{\gamma}}(\boldsymbol{\varepsilon}_{\ell}^{\boldsymbol{W}}), \mathcal{S}\right) + \sum_{\mu_i, \sigma_i}(\mu_i + \sigma_i), \tag{12}$$

where $L$ is the number of Monte-Carlo samples $\boldsymbol{\varepsilon}_1, \ldots, \boldsymbol{\varepsilon}_L$. Note that the first term of (12) is the cost function for the MAP problem (8) evaluated at $\{\boldsymbol{\mu}, \boldsymbol{W}\} = \{g_{\boldsymbol{\gamma}}(\boldsymbol{\varepsilon}_{\ell}^{\boldsymbol{\mu}}), g_{\boldsymbol{\gamma}}(\boldsymbol{\varepsilon}_{\ell}^{\boldsymbol{W}})\}$ for $\ell \in [n]$. Hence, the E-step summarizes into maximizing the right-hand side of (12) with respect to $\boldsymbol{\gamma}$ using gradient descent.

In the M-step, the ELBO is used as a proxy for the evidence $p_{\alpha}(\mathcal{S})$ and is maximized with respect to the hyper-parameters $\boldsymbol{\alpha}$. Again, we rely on the re-parameterization technique and compute the unbiased estimate of the ELBO in (12). The maximum of the estimate (12) with respect to $\boldsymbol{\alpha}$ has a closed form that depends on the choice of prior. We provide the closed-form solutions in Appendix A for a few proposed priors that emulate common regularizers. To avoid fast changes in $\boldsymbol{\alpha}$ due to the variance of the Monte-Carlo integration, we take an update similar to the one in [6] and take a weighted average between the current estimate and the minimizer of the current Monte-Carlo estimate of the ELBO as

$$\boldsymbol{\alpha} := \beta \cdot \boldsymbol{\alpha} + (1 - \beta) \cdot \underset{\tilde{\boldsymbol{\alpha}}}{\text{argmin}} \, \frac{1}{L} \sum_{\ell=1}^{L} \log p_{\tilde{\boldsymbol{\alpha}}}\left(g_{\boldsymbol{\gamma}}(\boldsymbol{\varepsilon}_{\ell}^{\boldsymbol{\mu}}), g_{\boldsymbol{\gamma}}(\boldsymbol{\varepsilon}_{\ell}^{\boldsymbol{W}}), \mathcal{S}\right), \tag{13}$$

where $\beta \in [0, 1]$ is the momentum term.

Algorithm 1 summarizes the proposed variational EM approach.[3] The computational complexity of the inner-most loop of Algorithm 1 is $L$ times the complexity of an iteration of gradient descent on the log-likelihood. However, as observed by recent studies in variational inference, using $L = 1$ is usually sufficient in many applications [18]. Hence, we use $L = 1$ in all our experiments, leading to the same computational complexity per-iteration as MLE using gradient descent.

**Algorithm 1** Variational EM algorithm for Multivariate Hawkes Processes

---

**Input:** Sequence of observations $\mathcal{S} = \{(t_n, i_n)\}_{n=1}^N$. Initial values for $\boldsymbol{\alpha}$ and $\boldsymbol{\gamma}$. Momentum term $0 \leq \beta < 1$. Sample size $L$ of Monte-Carlo integrations. Number of iterations $T_E$ and $T_{EM}$ of E-steps and EM-steps. Learning rate $\eta$.

1: **for** $t \leftarrow 1, \ldots, T_{EM}$ **do**
2:     **for** $t \leftarrow 1, \ldots, T_E$ **do**                                            ▷ E step
3:         Sample Gaussian noise $\boldsymbol{\varepsilon}_1, \ldots, \boldsymbol{\varepsilon}_L \sim \mathcal{N}(0, I)$.
4:         Evaluate the ELBO using Equation (12)
5:         Update $\boldsymbol{\nu} \leftarrow \boldsymbol{\nu} + \eta(\nabla_{\boldsymbol{\nu}} f(\boldsymbol{\nu}, \boldsymbol{\sigma}, \boldsymbol{\varepsilon}; \boldsymbol{\alpha}) + \mathbf{1})$.
6:         Update $\boldsymbol{\sigma} \leftarrow \boldsymbol{\sigma} + \eta(\nabla_{\boldsymbol{\sigma}} f(\boldsymbol{\nu}, \boldsymbol{\sigma}, \boldsymbol{\varepsilon}; \boldsymbol{\alpha}) + \mathbf{1})$.
7:     **end for**
8:     Sample $L$ Gaussian noise $\boldsymbol{\varepsilon}_1, \ldots, \boldsymbol{\varepsilon}_L$.                            ▷ M step
9:     Update $\boldsymbol{\alpha}$ using Equation (13).
10: **end for**
**Output:** $\boldsymbol{\alpha}, \boldsymbol{\gamma}$

---

# 5 Experimental Results

We carry out two sets of experiments. First, we perform a link-prediction task on synthetic data to show that our approach can accurately recover the support of the excitation matrix of the MHP under short sequences. Second, we perform an event-prediction task on real datasets of short sequences to show that our approach outperforms state-of-the-art methods in terms of predictive log-likelihood.

We run our experiments in two different settings. First, in a *parametric* setting where the exponential form of the excitation function is known, we compare our approach (VI-EXP) to the state-of-the-art MLE-based method (MLE-ADM4) from Zhou et al. [31]. Second, we use a *non-parametric* setting where no assumption is made on the shape of the excitation function. We then set the excitation function as a mixture of $M = 10$ Gaussian kernels defined as

$$\kappa_m(t) = (2\pi b^2)^{-1} \exp\left(-(t - \tau_m)^2/(2b^2)\right), \forall m = 1, \ldots, M, \tag{14}$$

where $\tau_m$ and $b$ are the known location and scale of the kernel. In this setting, we compare our approach (VI-SG) to the state-of-the-art MLE-based methods (MLE-SGLP) of Xu et al. [28] with the same $\{\kappa_m(t)\}$ [4]. Let us stress that the parametric methods have a strong advantage over the non-parametric ones because they are given the true value of the exponential decay $\zeta$.

As our VI approach returns a posterior on the parameters, rather than a point estimate, we use the mode of the approximate log-normal posterior as the inferred edges $\{\hat{w}_{ij}\}$. For the non-parametric setting, we use $\hat{w}_{ij} = \sum_{m=1}^M \hat{w}_{ij}^m$. To mimic the regularization schemes of the baselines, we use a Laplacian prior for the edge weights $\{w_{ij}\}$ to enforce sparsity, and we use a Gaussian prior for the baselines $\{\mu_i\}$. We tune the hyper-parameters of the baselines using grid search[5].

## 5.1 Synthetic Data

We first evaluate the performance of our VI approach on simulated data. We generate random Erdős–Rényi graphs with $D = 50$ nodes and edge probability $p = \log(D)/D$. Then, a sequence of observations is generated from an MHP with exponential excitation functions defined in (2) with exponential decay $\zeta = 1$. The baselines $\{\mu_i^*\}$ are sampled independently in $\mathrm{Unif}[0, 0.02]$, and the edge weights $\{w_{ij}^*\}$ are sampled independently in $\mathrm{Unif}[0.1, 0.2]$. Results are averaged over 30 graphs with 10 simulations each. For reproducibility, a detailed description of the experimental setup is provided in Appendix E.

To investigate if the support of the excitation matrix can be accurately recovered under small data, we evaluate the performance of each approach on three metrics [32, 28, 14].

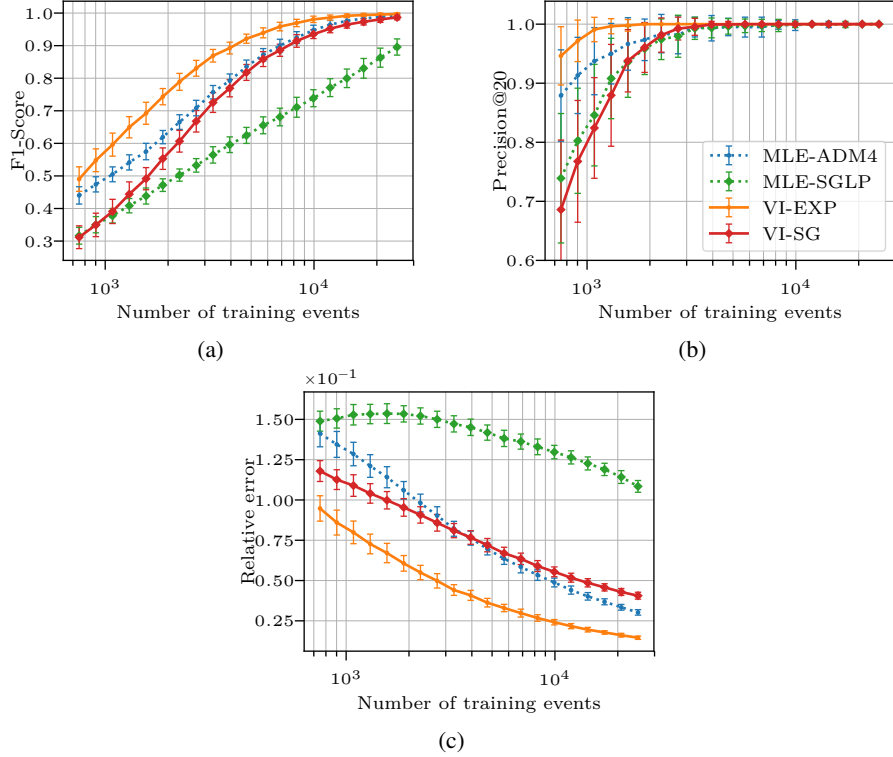

Figure 1: Performance measured by (a) F1-Score, (b) Precision@20, and (c) Relative error with respect to the number of training samples. Our VI approaches are shown in solid lines. The non-parametric methods are highlighted with square markers. Results are averaged over 30 random graphs with 10 simulations each ($\pm$ standard deviation).

- **F1-score.** We zero-out small weights using a threshold $\eta = 0.04$ and measure the F1-score of the resulting binary edge classification problem[6].

- **Precision@k.** Instead of thresholding, we also report the precision@k defined by the average fraction of correctly identified edges in the top $k$ largest estimated weights. Since the proposed VI approach gives an estimate of uncertainty via the variance of the posterior, we select the edges with high weights $\hat{w}_{ij}$ and low uncertainty, *i.e.*, the edges with ratio of lowest standard deviation over weight $\hat{w}_{ij}$.

- **Relative error.** To evaluate the distance of the estimated weights to the ground truth ones, we use the averaged relative error defined as $|\hat{w}_{ij} - w^*_{ij}|/w^*_{ij}$ when $w^*_{ij} \neq 0$, and $\hat{w}_{ij}/(\min_{w^*_{kl}>0} w^*_{kl})$ when $w^*_{ij} = 0$. This metric is more sensitive to errors in small weights $w^*_{ij}$, and therefore penalizes false positive over false negative errors.

We investigate the sensitivity of each approach to the amount of data available for training by varying the size of the training set from $N = 750$ to $N = 25\,000$ events, *i.e.*, 15 to 500 events per node. Results are shown in Figure 1. Our approach improves the results in both parametric and non-parametric settings for all metrics. The improvements are more substantial in the non-parametric setting. If the accuracy of the top edges is similar for both VI-SG and MLE-SGLP in terms of precision@20, VI-SG improves the F1-score by about 20% with $N = 5\,000$ training events. The reason for this improvement is that MLE-SGLP has a much higher false positive rate, which is hurting the F1-score but does not affect the precision@20. VI-SG is also able to reach the same F1-score as the parametric baseline MLE-ADM4 with only $N = 4\,000$ training events[7]. Note that VI-SG is optimizing $D^2M + D = 25\,050$ hyper-parameters with minimal additional cost.

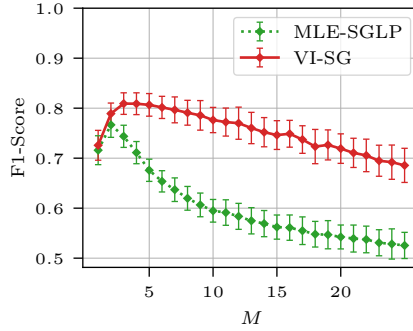

Figure 2: Analysis of the robustness of non-parametric approaches to the number of bases $M$ of excitation functions (for fixed $N = 2\,000$).

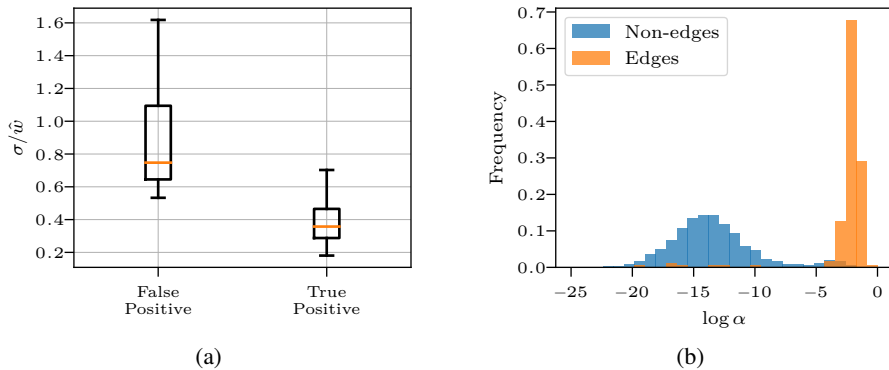

Figure 3: Analysis of the uncertainty of the parameters learned by VI-EXP (for fixed $N = 5\,000$). (a) Uncertainty of the inferred edges and (b) histogram of learned $\alpha$. The learned $\alpha$ are smaller for non-edges, and false positive edges have higher uncertainty than the true positive ones.

In the next experiment, we focus on the non-parametric setting, we fix the length of observation to $N = 5000$ and study the effect of increasing $M$ on the performance of the algorithms. The results are shown in Figure 2. We see that our approach is more robust to the choice of $M$ than MLE-SGLP. A possible explanation for this behavior is that MLE-SGLP overfits due to the increasing number of model parameters.

Finally, we investigate the parameters of the model learned by our VI-EXP approach. In Figure 3a, we use the variance of the approximated posterior $q_\gamma$ as a measure of confidence for edge identification, and we report the distribution of ratio of standard deviation over weight $\hat{w}_{ij}$ for both the true and false positive edges. Similar results hold between the true and false negative edges. The false positive edges have a higher uncertainty than the true positive ones. This is relevant when we cannot identify all edges due to lack of data, even though we still wish to identify a subset of edges with high confidence. In addition, Figure 3b confirms that, as expected, the optimized weight priors $\alpha$ are much larger for true edges in the ground-truth excitation matrix than for non-edges.

## 5.2 Real Data

We also evaluate the performance of our approach on the following three small datasets:

1. **Epidemics.** This dataset contains records of infection of individuals, along with their corresponding district of residence, during the last Ebola epidemic in West Africa in 2014-2015 [15]. To learn the propagation network of the epidemics, we consider the 54 districts as processes and define infection records as events.

2. **Stock market.** This dataset contains the stock prices of 12 high-tech companies sampled every 2 minutes on the New York Stock Exchange for 20 days in April 2008 [13]. We consider each stock as a process and record an event every time a stock price changes by $0.15\%$ from its current value.

3. **Enron email.** This dataset contains emails between employees of Enron from the Enron corpus. We consider all employees with more than 10 received emails as processes and record an event every time an employee receives an email.

We perform an event-prediction task to show that our approach outperforms the state-of-the-art methods in terms of predictive log-likelihood. To do so, we use the first 70% events as training set, and we compute the held-out averaged log-likelihood on the remaining 30%. We present the results in Table 1.

We first see that the non-parametric methods outperform the parametric ones on both the Epidemic dataset and the Stock market dataset. This suggests that the exponential excitation function might be too restrictive to fit their excitation patterns. In addition, our non-parametric approach VI-SG significantly outperforms MLE-SGLP on all datasets. The improvement is particularly clear for the Epidemic dataset, which has the smallest number of events per dimension. Indeed, the top edges learned by VI-SG correspond to contiguous districts as expected. This is not the case for MLE-SGLP, for which the top learned edges correspond to districts that are far from each other.

Table 1: Predictive log-likelihood for the models learned on various real datasets.

| Dataset | Statistics | | Averaged predictive log-likelihood | | | |
|---------|-----------|-----------|--------|-----------|--------|----------|
| | #dim ($D$) | #events ($N$) | **VI-SG** | MLE-SGLP | **VI-EXP** | MLE-ADM4 |
| Epidemics | 54 | 5 349 | $-\mathbf{2,06}$ | $-3,03$ | $-4,31$ | $-4,61$ |
| Stock market | 12 | 7 089 | $-\mathbf{1,00}$ | $-2,45$ | $-2,82$ | $-2,81$ |
| Enron email | 143 | 74 294 | $-0,42$ | $-1,01$ | $-\mathbf{0,23}$ | $-0,40$ |

# 6   Conclusion

We proposed a novel approach to learn the excitation matrix of a multivariate Hawkes process in the presence of short observation sequences. We observed that state-of-the-art methods are sensitive to the amount of data used for training and showed that the proposed approach outperforms these methods when only short training sequences are available. The common tool to tackle this problem is to design smarter regularization schemes. However, all maximum likelihood-based methods suffer from a common problem: all the model parameters are regularized equally with a few hyper-parameters. We developed a variational expectation maximization algorithm that is able to (1) optimize over an extended set of hyper-parameters, with almost no additional cost and (2) take into account the uncertainty of the learned model parameters by fitting a posterior distribution over them. We performed experiments on both synthetic and real datasets and showed that our approach outperforms state-of-the-art methods under small-data regimes.

## Acknowledgments

We would like to thank Negar Kiyavash and Jalal Etesami for the valuable discussions and insightful feedback at the early stage of this work. The work presented in this paper was supported in part by the Swiss National Science Foundation under grant number 200021-182407.

## Footnotes

[2]This assumption can be relaxed using more advanced techniques at the cost of having a higher computational complexity.

[3]Source code is available publicly.

[4] We also performed the experiments with other approaches designed for large-scale datasets, but their performance was below that of the reported baselines [1, 20, 21].

[5] More details are provided in Appendix E

[6]Additional results with varying thresholds $\eta$ are provided in Appendix D.

[7]We present additional results with various thresholds $\eta$ and $k$ in Appendix D.

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
