[Supplementary Material]

# A  Different Priors

In this section, we provide the probabilistic interpretation as a prior of several regularizers commonly used in the literature.

- $L_2$**-regularizer:** Perhaps the most commonly used regularizer in MHPs is the $L_2$-regularizer $w_{ij}^2/(2\alpha)$. $L_2$-regularizer can be interpreted as a zero-mean Gaussian distribution over the weights, *i.e.*,

$$p_\alpha(w_{ij}) = \frac{1}{\sqrt{2\pi\alpha}} \exp(-\frac{w_{ij}^2}{2\alpha}),$$

  where $\alpha$ is the variance.

- $L_1$**-regularizer:** This regularizer (also known as lasso regularizer) is considered as a convex surrogate for the $L_0$ (pseudo) norm. Hence, it promotes the sparsity of the parameters. It can be interpreted as a Laplace distribution over the weights, *i.e.*,

$$p_\alpha(w_{ij}) = \frac{1}{2\alpha} \exp(-\frac{|w_{ij}|}{\alpha}).$$

- **Low-rank regularizer:** To achieve a low-rank excitation matrix $\boldsymbol{W}$, a nuclear norm penalty on $\boldsymbol{W}$ is often used as a regularizer [31], thus enabling clustering structures in $\boldsymbol{W}$. For an excitation matrix $\boldsymbol{W} \in \mathbb{R}^{D \times D}$, let $\boldsymbol{w}_{\cdot,j} = [w_{1,j} \ldots, w_{D,j}]$, then the different $\{\boldsymbol{w}_{\cdot,j}\}_j$ are independent for different $j$ and the prior over $\boldsymbol{w}_{\cdot,j}$ is

$$p_\alpha(\boldsymbol{w}_{\cdot,j}) = c \cdot \frac{1}{\alpha^D} \exp(-\frac{\|\boldsymbol{w}_{\cdot,j}\|_2}{\alpha}),$$

  where $c > 0$ is a constant.

- **Group-lasso regularizer:** This regularizer is used in [28] in the non-parametric setting defined in Section 3 where the excitation function is approximated by a linear combination of $M$ basis functions, parameterized by $\boldsymbol{w}_{ij} = [w_{ij}^1, \ldots, w_{ij}^M]$. In this case, the $L_2$-norm of $\boldsymbol{w}_{ij}$ is assumed to have a Laplace distribution, *i.e.*,

$$p_\alpha(\boldsymbol{w}_{ij}) = c \cdot \frac{1}{\alpha^M} \exp(-\frac{\|\boldsymbol{w}_{ij}\|_2}{\alpha}),$$

  where $c > 0$ is a constant.

# B  Hyper-parameter Update

Below, we give a closed-form solution to update $\boldsymbol{\alpha}$ in (13) for two priors used in the experiments of Section 5. The update rules for the other priors are similar. We start by rewriting the joint distribution $\log p_{\boldsymbol{\alpha}}(g_{\boldsymbol{\gamma}}(\boldsymbol{\varepsilon}_\ell), \mathcal{S})$ as

$$\log p_{\boldsymbol{\alpha}}(g_{\boldsymbol{\gamma}}(\boldsymbol{\varepsilon}_\ell), \mathcal{S}) = \log p(\mathcal{S}|g_{\boldsymbol{\gamma}}(\boldsymbol{\varepsilon}_\ell)) + \log p_{\boldsymbol{\alpha}}(g_{\boldsymbol{\gamma}}(\boldsymbol{\varepsilon}_\ell)), \tag{15}$$

where the first term (the likelihood) is not a function of $\boldsymbol{\alpha}$ and only the second term (the prior) is a function of $\boldsymbol{\alpha}$. Hence minimizing $\sum_{\ell=1}^n \log p_{\boldsymbol{\alpha}}(g_{\boldsymbol{\gamma}}(\boldsymbol{\varepsilon}_\ell), \mathcal{S})$ over $\boldsymbol{\alpha}$ amounts to minimizing $\sum_{\ell=1}^n \log p_{\boldsymbol{\alpha}}(g_{\boldsymbol{\gamma}}(\boldsymbol{\varepsilon}_\ell))$.

$L_2$**-regularizer:**  For the $L_2$-regularizer we have

$$\log p_\alpha(g_{\boldsymbol{\gamma}}(\boldsymbol{\varepsilon}_\ell)) = -\frac{g_{\boldsymbol{\gamma}}(\boldsymbol{\varepsilon}_\ell)^2}{2\alpha} - \frac{1}{2} \log \alpha - \frac{1}{2} \log 2\pi,$$

and the closed form update hence becomes

$$\underset{\tilde{\alpha}}{\mathrm{argmin}} \frac{1}{n} \sum_{\ell=1}^n \log p_{\tilde{\alpha}}(g_{\boldsymbol{\gamma}}(\boldsymbol{\varepsilon}_\ell), \mathcal{S}) = \frac{1}{n} \sum_{\ell=1}^n g_{\boldsymbol{\gamma}}(\boldsymbol{\varepsilon}_\ell)^2. \tag{16}$$

$L_1$**-regularizer:** For the $L_1$-regularizer we have

$$\log p_\alpha(g_\gamma(\boldsymbol{\varepsilon}_\ell)) = -\frac{g_\gamma(\boldsymbol{\varepsilon}_\ell)}{\alpha} - \log \alpha - \log 2,$$

and the closed form update hence becomes

$$\operatorname*{argmin}_{\tilde{\alpha}} \frac{1}{n}\sum_{\ell=1}^{n} \log p_{\tilde{\alpha}}(g_\gamma(\boldsymbol{\varepsilon}_\ell),\mathcal{S}) = \frac{1}{n}\sum_{\ell=1}^{n} g_\gamma(\boldsymbol{\varepsilon}_\ell). \tag{17}$$

## C  Simple Optimization of $\alpha$

In this section, we show that we cannot simply find $\boldsymbol{\alpha}$ by optimizing the negative log-likelihood in (4) or the MAP objective in (8) over $\boldsymbol{\alpha}$.

Fist note that, minimizing regularized negative log-likelihood in (4) over $\boldsymbol{\alpha}$, simply sets $\boldsymbol{\alpha}$ to infinity.

Second, we show that maximizing the MAP objective in (8) over $\boldsymbol{\alpha}$ also fails because it is unbounded from above. We show this for the case of the Gaussian prior defined by

$$p_{\boldsymbol{\alpha}}(\boldsymbol{\mu},\boldsymbol{W}) = p_{\boldsymbol{\alpha}^\mu}(\boldsymbol{\mu})p_{\boldsymbol{\alpha}^W}(\boldsymbol{W}) = \frac{1}{\sqrt{2\pi\boldsymbol{\alpha}^\mu}}\exp\left(-\frac{\|\boldsymbol{\mu}\|^2}{2\boldsymbol{\alpha}^\mu}\right)\cdot\frac{1}{\sqrt{2\pi\boldsymbol{\alpha}^W}}\exp\left(-\frac{\|\boldsymbol{W}\|^2}{2\boldsymbol{\alpha}^W}\right). \tag{18}$$

but the same result holds for other priors. The log of the Gaussian prior (18) is

$$\begin{aligned}
\log p_{\boldsymbol{\alpha}}(\boldsymbol{\mu},\boldsymbol{W}) &= \log p_{\boldsymbol{\alpha}^\mu}(\boldsymbol{\mu}) + \log p_{\boldsymbol{\alpha}^W}(\boldsymbol{W})\\
&= -\frac{\|\boldsymbol{\mu}\|^2}{2\boldsymbol{\alpha}^\mu} - \frac{\|\boldsymbol{W}\|^2}{2\boldsymbol{\alpha}^W} - \frac{1}{2}\log\boldsymbol{\alpha}^\mu - \frac{1}{2}\log\boldsymbol{\alpha}^W + c,
\end{aligned} \tag{19}$$

where $c$ is a constant independent of $\boldsymbol{\alpha}$. In the MAP objective (8), if we set $\boldsymbol{\mu} = 1$ and $\boldsymbol{W} = 0$, *i.e.*, all processes are simple Poisson process with rate 1 and no interaction between them, then the conditional intensity $\lambda_i(t) = 1$ for all $i \in [d]$ and $t \geq 0$. The log-likelihood in (5) becomes $\log p(\mathcal{S}|\boldsymbol{\mu},\boldsymbol{W}) = -DT$, which is bounded from below. Set $\boldsymbol{\alpha}^\mu = 1$, then for $\boldsymbol{\alpha}^W \to 0^+$, we get $\log p_{\boldsymbol{\alpha}}(\boldsymbol{\mu},\boldsymbol{W}) \to \infty$. Hence, the MAP estimator for $\boldsymbol{\alpha}$ is unbounded from above and maximizing the MAP objective simultaneously over both the hyper-parameters $\boldsymbol{\alpha}$ and the model parameters $\boldsymbol{\mu}$ and $\boldsymbol{W}$ would fail.

## D  Additional Experimental Results

We first carry out an additional set of experiments to show the effect of the zeroing-out small weights using a threshold $\eta$. To do so, we first need to introduce the following two performance metrics:

- The *false positive rate* (FPR) to be the fraction of errors in learnt edges

$$\mathrm{FPR} = |\{\hat{w}_{ij}|\hat{w}_{ij} > 0, w_{ij}^\star = 0\}|/|\{\hat{w}_{ij}|w_{ij}^\star = 0\}|,$$

 where $|\cdot|$ denotes the cardinality of a set.
- Similarly, the *false negative rate* (FNR) to be the fraction of errors in learnt non-edges

$$\mathrm{FNR} = |\{\hat{w}_{ij}|\hat{w}_{ij} = 0, w_{ij}^\star > 0\}|\,|\{\hat{w}_{ij}|w_{ij}^\star > 0\}|.$$

Figure 4 shows the effect of number of samples on F1-score for several choice of threshold $\eta$. We see that our proposed algorithm VI-EXP (resp. VI-SG) outperform its MLE counterpart MLE-ADM4 (resp. MLE-SGLP) for all values of $\eta$. With increasing $\eta$, we see that the F1-score of MLE-based approaches improve. This is due to the FPR decreasing faster than the FNR increases due to the sparsity of the graph. However note that, since we do not know the expected value of true edges $w_{ij}^*$ beforehand, it is not clear a-priori what value we should set for the threshold $\eta$. Ideally, we choose the threshold $\eta$ to be as small as possible, which is the regime in which our variational inference algorithm outperforms MLE-based methods the most.

In Figure 5, we plot Precision@$k$ for different values of $K$. The number of edges in the generated synthetic graphs is 195, so in Figure 5 we vary $K$ up to 195. We see that VI-EXP always has better

Figure 4: Performance measured by F1-Score with respect to the number of training samples. The proposed variational inference approaches are shown in solid lines. The non-parametric methods are highlighted with square markers.

Figure 5: Performance measured by Precision@$k$ with respect to the number of training samples. The proposed variational inference approaches are shown in solid lines. The non-parametric methods are highlighted with square markers.

Precision@$k$ than its counterpart MLE-ADM4. VI-SG has the same Precision@$k$ for $k = 5$, 10, and 20 as MLE-SGLP. For larger $k$ MLE-SGLP has slightly better Precision@$k$. Note that, Precision@$k$ focuses only on the accuracy of top $k$ edges learnt by an algorithm and hence does not discriminate the imbalance between precision and recall for large $k$ in sparse graphs.

Finally, to evaluate the scalability of our approach, we ran additional simulations on increasingly large-dimensional problems. As shown in Figure 6, the per-iteration running time of our approach VI-EXP (implemented in python) scales better than the one of MLE-ADM4 (implemented in C++). In addition, even if our gradient descent algorithm requires more iterations to converge, we show in Figure 7 that VI-EXP reaches the same F1-score as MLE-ADM4 faster.

Figure 6: Comparison of running time per-iteration.

Figure 7: Running time required for our approach VI-EXP to reach the same F1-Score as MLE-ADM4.

# E  Reproducibility

In this section, we provide extensive details on the experimental setup used in Section 5. We first describe the implementation details of the algorithm described in Algorithm 1. We then provide the details of the experimental setup for both the synthetic and real data experiments.

## E.1  Implementation details of Algorithm 1

We used $L = 1$ sampled Gaussian noise in line 3 of Algorithm 1. We set the momentum term $\beta = 0.5$ in (13). In our early experiments, we observed that the performance of the algorithm is not sensitive to the momentum term $\beta$ for $\beta \in (0, 1)$. Therefore, we decided to set it to $0.5$ in all experiments. We used the Adam optimizer with learning rate $\eta = 0.02$. We also multiply the learning rate by $1 - 10^{-4}$ at each iteration. Both $\nu^{\mu}$ and $\nu^{W}$ were initialized by sampling from the normal distribution $N(0.1, 0.01)$. We initialized $\alpha = 0.1$ for all hyper-parameters. We observed that the performance of the algorithm is not sensitive to the initialization. Both $\sigma^{\mu}$ and $\sigma^{W}$ were initialized by sampling from the normal distribution $N(0.2, 0.01)$ then clipping them to be in $[0.01, 2]$. This initialization ensures that the initial variance of the algorithm is neither small nor too big.

## E.2  Synthetic experiments

To create the synthetic data, we generated random Erdős–Rényi graphs with $D = 50$ nodes and with edge probability $p = \log(D)/D$, leading to graphs with 195 edges on average. Then, the sequences of observations were generated from an MHP with the exponential excitation kernel defined in (2). The baseline $\{\mu_i\}$ were sampled uniformly at random in $[0, 0.02]$, and the edge weights $\{w_{ij}^*\}$ were sampled uniformly at random in $[0.1, 0.2]$. To generate the results of Figure 1, we varied the length of observations between $N = 700$ and $N = 25000$. The results were averaged over 30 graphs with 10 simulations each.

We used `tick`[8] Python library to run the MLE-based baseline approaches. To tune the hyper-parameters of the MLE-based approaches, we first manually searched for an initial range of parameters where the algorithm performed well. Then, we fined-tuned the hyper-parameters using grid-search to find the ones giving the best results for the Precision@20 and F1-score metrics. For MLE-SGLP, we used the grid range $1/\alpha \in [0.001, 0.005, 0.01, 0.05, 0.1, 0.5, 1.0]$ and `lasso_grouplasso_ratio` $\in [0.25, 0.5, 0.75]$. We used the default values for the optimizer, which we checked and are sure of its convergence. We finally chose $1/\alpha = 0.1$ and `lasso_grouplasso_ratio` $= 0.75$. For MLE-ADM4, we also used the grid range $1/\alpha \in [0.001, 0.005, 0.01, 0.05, 0.1, 0.5, 1.0]$ and lasso_nuclear_norm $\in [0.25, 0.5, 0.75]$. Making overall 21 different configurations. We finally chose $1/\alpha = 0.05$ and lasso_nuclear_norm $= 0.5$ that gave the best results for Precision@20 and F1-score.

## E.3  Real data experiments

For our approach VI-EXP and its parametric counterpart VI-SG, the exponential decay parameter must be tuned for each dataset. As expected, both algorithms performed best with the same decay.

For our approach VI-SG and its MLE counterpart MLE-SGLP, there are two parameters to tune, $M$ and cutoff time $T_c$. The center of the $m$-th Gaussian kernel, with $m \in [M]$, is defined as $t_m = T_c \cdot (m - 1)/M$ and its scale is defined as $b = T_c/(\pi \cdot M)$ in (3). After manually finding an initial range of $M$ and $T$ where algorithms performed well, we then fine-tuned them using the grid-search.

**Epidemic dataset.**   For our VI-SG algorithm, we did a grid-search with $M \in [30, 35, 40, 45, 50, 55]$ and $T_c \in [025 \cdot M, 0.5 \cdot M, 0.75 \cdot M]$. We did not see a notable difference between the performance of different grids, as long as $M$ and $T$ are large enough. We chose $M = 55$ and $T = 27.5$. For the baseline MLE-SGLP, we did a grid-search with $M \in [1, 3, 5, 7, 9, 11, 13, 15, 17, 19, 21]$, $T_c \in [0.25M, 0.5M, 1M, 2M, 5M, 10M, 20M, 40M]$ and $1/\alpha \in [1, 10, 50, 100]$, that makes overall 352 experiments. We chose $M = 19$, $T_c = 9.5$ and $1/\alpha = 10$. For our algorithm VI-EXP, we tried decay $\in [0.1, 0.5, 1, 2, 5, 10, 20, 40]$ and we chose decay $= 0.1$. For the baseline MLE-ADM4, we did a grid-search with decay $\in [0.1, 0.5, 1, 2, 5, 10, 20, 40]$ and $1/\alpha = [0.01, 0.1, 1, 2, 5, 10, 50, 100, 200, 400, 800]$. We chose decay $= 0.1$ and $1/\alpha = 50$.

**Stock market dataset.**   In the stock market dataset, our algorithm VI-SG also performed better with a larger $M$. As for large $M$ the experiments are slow we decided to set $M = 50$ and did grid-search for $T_c$ with $T_c \in [0.15 \cdot M, 0.25 \cdot M, 0.5 \cdot M]$. For the baseline MLE-SGLP, we did a grid-search with $M = [1, 3, 5, 7, 9, 11, 13, 15, 17, 19, 21]$, $T_c \in [0.25 \cdot M, 0.5 \cdot M, 0.75 \cdot M, 1 \cdot M, 2 \cdot M, 5 \cdot M]$ and $1/\alpha \in [0.01, 0.1, 0.5, 1, 10, 50, 100]$. The best values found were $M = 17$, $T_c = 8.5$ and $C = 0.1$. For our algorithm VI-EXP, we tried decay $\in [0.1, 0.5, 1, 2, 5, 10, 20, 40]$ and we chose decay $= 0,1$. For the baseline MLE-ADM4, we did a grid search with decay $\in [0.1, 0.5, 1, 2, 5, 10, 20, 40]$ and $1/\alpha = [0.01, 0.1, 1, 2, 5, 10, 50, 100, 200, 400, 800]$. We chose decay $= 0,1$ and $1/\alpha = 1$.

**Enron email dataset.**   The Enron email dataset is a larger dataset and experiments are more computationally intensive, so we chose smaller ranges for hyper-parameter tuning. For our algorithm VI-SG we did a grid-search with $M = 10$ and $T_c \in [5, 7.5, 10, 15]$. The best value is $T_c = 5$. For the baseline MLE-SGLP, we did a grid-search with $M = [1, 2, 3, 4, 5]$, $T_c \in [0.1, 0.25, 0.5, 0.75, 1, 1.25]$ and $1/\alpha \in [10, 20, 50, 100, 500]$. The best value is $M = 1$, $T_c = 2.5$ and $1/\alpha = 50$. For our algorithm VI-EXP, we tried decay $\in [5, 10, 20, 40]$ and we chose decay $= 20$. For the baseline MLE-ADM4, we did a grid-search with decay $\in [0.1, 0.5, 1, 2, 5, 10, 20, 40]$ and $1/\alpha = [0.01, 0.1, 1, 2, 5, 10, 50, 100, 200, 400, 800]$. We chose decay $= 20$ and $1/\alpha = 0.1$.

## Footnotes

[8]https://github.com/X-DataInitiative/tick