[Reviews · NeurIPS 2019]

Reviewer 1



The authors present a variational EM algorithm for inference in a multivariate Hawkes process. This allows learning in the setting with many hyper-parameters and limited data, where ML approaches will fail or at least hyper-parameter estimation would be difficult. ML approaches therefore tend to use less hyper-parameters which may limit their performance. This class of model has seen a lot of interest recently and I’m not aware of a previous approach to variational inference in the continuous-time setting. A previous stochastic variational inference approach from Linderman and Adams (cited here) used a discrete-time approximation and other Bayesian approaches that I’m aware of are based on MCMC. This therefore looks like a timely and valuable contribution. Experiments are run to look at performance in terms of link and event prediction and for both parametric and non-parametric excitation function models. The VI approach is shown to outperform ML approaches on limited data and looks better even for pretty large datasets according to the synthetic data evaluation. VI also seems to deal better with an over-complex non-parametric function compared to ML. Results on three real datasets also look very good compared to MLE approaches.

Reviewer 2



Post rebuttal: Many thanks for addressing my concerns. I have read the rebuttal and other reviewer's comments. Please elaborate them in more details as needed in the revision. 1. Regarding (8) -> (9) transformation: perhaps there could be a data-dependent way of learning a better prior of \alpha. 2. Regarding my comment that the case where one dimension has scarce data. The scarcity of the data is caused by the naturally smaller intensity function. If one dimension has a harder to learn triggering function, it may propagate onto other dimensions. In prior works, such cases are hard to handle, and I think the method proposed here could be used as a remedy to this issue. I don't think the intuition explanation is a satisfying answer and I believe some additional exploration is warranted here. Overall I maintain my vote to accept the paper. -------------------------------------------------------------------------------------- This paper proposes learning the parametric MHP by regularized MLE. The parameters to be learned are \mu and W, and a limitation of the traditional algorithms are that all the dimensions share the same regularization parameter. The authors proposed a Bayesian method and modeled \alpha, the regularization coefficient, as a hidden variable. An EM algorithm follows naturally. Experimental results show that the proposed method improves upon the EM-based learning algorithm MLE-SGLP proposed by Xu et al. I feel like this is a good submission but it's quite possible I didn't understand the main part of the paper. I think the idea is novel and empirical evidence suggests it would help researchers to select the regularization parameters. In cases where learning samples is scarce, I think this is particularly useful. Empirical evidence suggests that the method works, but formulation-wise, there are a few places that confused me although I already think it's quite interesting. 1. The first key step in the paper is from (4) to (8), where the regularizer is viewed as \log p_{\alpha}(\mu,W). I would argue that the ignored normalization coefficient here depends on \mu and W itself, and cannot be omitted. Or \alpha has to depend on \mu and W (which is not the case per Appendix). Therefore I suppose this is not an equivalent transformation? I also think it is this reason that leads to the fact that one cannot optimize \alpha in (4) directly, am I right for this? I didn't understand why one can optimize over (8) (stated in L142) as one can change \alpha arbitrarily and it doesn't affect the choice of the parameters which is determined by the problem. 2. I didn't quite understand the intuition behind the transformation from (8) to (9). In L147 I suppose the upper bound should be the maximum likelihood instead of the likelihood? 3. I saw in appendix several choices of p_{\alpha}. How did you obtain this from (4)? Any reasons why p_{\alpha} should be designed as described in Appendix? Understanding those along with other reviewer's concerns would help me better appreciate the result of this paper.

Reviewer 3



In the inference problem of multivariate Hawkes processes, the regularization is critical for avoiding over-fitting when only short sequences of events are available. This paper develops the variational EM algorithm, which allows for efficiently determining a lot of hyper-parameters that control the influence of regularization penalty. Also, it can estimate the model parameters while considering the uncertainty of them. This paper is well-written and easily understandable. The proposed algorithm is practically useful but seems to be a straightforward application of the existing techniques. Comments following the guidelines as requested. [Originality] Strengths. The authors make use of the black-box variational inference to effectively determine the hyper-parameters for controlling the regularization penalty. Weaknesses. My concern is that the proposed algorithm seems somewhat incremental, as it is a combination of well-known techniques. I think It would be interesting to consider the case of more high dimensional data, that is, the number of nodes is large. In that case, the authors might need to stretch the algorithm so that the computational complexity can be reduced. [Quality] Strengths. I think this paper is technically sound, and the experimental section is well-organized. There is convincing experimental evidence, i.e., analysis of the uncertainty, in Figure 3. Weaknesses. The limitations of the proposed approach are not discussed. How is the efficiency and execution time of inference? Could the authors provide enough experimental results or theoretical analysis? Especially, in all the experiments, the authors consider the case of relatively low dimensional data (up to 143). It would be a good idea to discuss the case of applying to more high dimensional data; is it applicable in terms of computation costs? The number of nodes might be larger in typical applications such as social-network studies (e.g., [1]). Several questions are as follows: 1) Why don't you optimize the decay-parameters? 2) Could the authors visualize the estimated causal-networks? I would like to see the networks that are estimated by the MLE-based method and the proposed method, respectively. [Clarity] This paper is easy to understand and provides enough information to implement. Small suggestion: In the re-parameterization of $\mu$ and $W$, the superscript notation (e.g., $\epsilon^\mu$) might be confusing. It would be better to denote them as alternative notations, e.g., $\epsilon_\mu$. [Significance] The inference problem of multivariate Hawkes processes under short observation sequences is an important task, and this paper has a useful contribution, easy to implement. However, I worry whether an application to the large-scale data sets whose number of nodes is large is difficult in practice. I think it would be significant to address the case of high dimensional data and provide analysis of the estimated causal-structures from the large-scale graphs. [1] T. Iwata et al., Discovering Latent Influence in Online Social Activities via Shared Cascade Poisson Processes, KDD, pages 266–274, 2013. ------------------------------ After author feedback: I appreciate the responses to my questions. The new experimental results in the rebuttal is a welcome addition. On the other hand, I think It would be a good idea to add the comparisons of the estimated excitation matrix $W$ and clarify the effect of the reguralization. Also, I still think that the variational inference scheme derived by the authors is a combination of several ML techniques; its technical contribution is somewhat low. But I think the proposed algorithm is effective and has a good impact for ML community and practitioners. Overall, my score for the paper remains the same.

[Author Response · NeurIPS 2019]



We would like to thank the reviewers for their detailed feedback and insightful comments, we will incorporate the
suggested clarifications in the paper.

**Remarks for Reviewer 1.**    We will add the reference of the SVI approach of Linderman and Adams in the introduction.

**Remarks for Reviewer 2.**

• *On the normalization coefficient and prior $p_\alpha$ in (4)*: It is known that the regularized-MLE objective is equivalent to
MAP objective up to the constant normalization coefficient. The regularization term $R(\mu, W)/\alpha$ in (4) can be seen
as the negative of the logarithm of the unnormalized prior. So to derive the prior $p_\alpha(\mu, W)$ we only need to compute
the normalization term, which is the integral of $\exp\left(-R(\mu, W)/\alpha\right)$ over $\mu$ and $W$, and which therefore cannot be a
function of the integration variables $\mu$ and $W$.
• *On Optimizing $\alpha$ in (4) and (8) directly*: In line 142 we actually mean that the MAP estimator cannot be optimized
over $\alpha$. Indeed, as demonstrated in the example of Appendix B, the MAP objective is an unbounded function (from
above) of $\alpha$.
• *On line 147*: Indeed it should be "maximum likelihood", we apologize for the confusion caused by the typo.
• *On performance if only 1 dimension has few observations*: In our setting, data scarcity comes from the short length
of the observation window and not from missing data. So, if one dimension $i$ has much fewer timestamps than
others, it means that overall, it has smaller intensity (i.e. small $\mu_i$ and incoming $W_{ij}$). Therefore the likelihood
function naturally enforces small values in these parameters to explain the observed intervals with no timestamps.
The setting where the data scarcity comes from both short observation window and missing data requires extending
our probabilistic model and is an interesting direction for future work.

**Remarks for Reviewer 3.**

• *Computational complexity of the algorithm*: Our gradient-based method is computationally efficient and scales well
to large data regimes. For small data-regimes, the state-of-the-art methods empirically seem to converge faster as
they need fewer iterations (even if there is no proof of convergence rate in the papers). However, when the number of
nodes gets large and the number of observations increases, the per-iteration cost of the state-of-the-art methods grows
faster than our gradient-based approach, which we expect to converge faster for such settings. Indeed, the complexity
of MLE-ADM4 is $O(N_{iter} n^3 d^2)$ (see Table 1, Achab et al. 2016, [1]), whereas the complexity of our approach can
be reduced to $O(N_{iter} n d^2 + d^2 n^2)$ where the reduced cost comes from efficiently pre-computing some constant
terms in the log-likelihood function (at the cost of memory), which is a one shot cost of $O(d^2 n^2)$.
• To evaluate the scalability of our approach, we ran additional simulations on increasingly large-dimensional problems.
As shown in Figure 1, the per-iteration running time of our approach VI-EXP (implemented in python) scales
better than the one of MLE-ADM4 (implemented in C++). In addition, even if our gradient descent algorithm
requires more iterations to converge, we show in Figure 2 that VI-EXP reaches the same F1-score as MLE-ADM4
faster. Empirically, we expect similar results for the non-parametric setting. We performed simulations only for the
parametric setting due to the time constraint of the rebuttal.
• *Optimizing the decay parameters*: It is possible to optimize the decay parameters but we chose to use this particular
form of exponential kernel as an example designed to match the efficient C++ implementation of MLE-ADM4,
which takes advantage of the convexity of the problem. In addition, considering a fixed decay enables the use of
caching as discussed above.
• *Visualizing the estimated causal-networks*: In high-dimensions, the resulting networks are not easy to visualize. We
tried to draw the learned networks on top of a map of the Ebola dataset, but the figure needs to be rendered too large
to be clear. Given space limitation, we did not plot any.

Figure 1: Comparison of per-iteration running time.

Figure 2: Running time required for our approach VI-EXP to reach the same F1-Score as MLE-ADM4.

[Meta-Review · NeurIPS 2019]

An overall nice contribution for fitting Hawkes processes. Clarification on optimization of decay parameters and per-iteration complexity would be great.